# Vocational Training in Virtual Reality: A Case Study Using the 4C/ID Model

Miriam Mulders 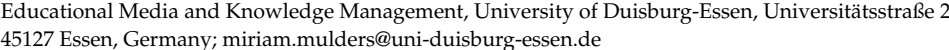

Educational Media and Knowledge Management, University of Duisburg-Essen, Universitätsstraße 2,
45127 Essen, Germany; miriam.mulders@uni-duisburg-essen.de

**Abstract:** Virtual reality (VR) is an emerging technology with a variety of potential benefits for vocational training. Therefore, this paper presents a VR training based on the highly validated 4C/ID model to train vocational competencies in the field of vehicle painting. The following 4C/ID components were designed using the associated 10 step approach: learning tasks, supportive information, procedural information, and part-task practice. The paper describes the instructional design process including an elaborated blueprint for a VR training application for aspiring vehicle painters. We explain the model's principles and features and their suitability for designing a VR vocational training that fosters integrated competence acquisition. Following the methodology of design-based research, several research methods (e.g., a target group analysis) and the ongoing development of prototypes enabled agile process structures. Results indicate that the 4C/ID model and the 10 step approach promote the instructional design process using VR in vocational training. Implementation and methodological issues that arose during the design process (e.g., limited time within VR) are adequately discussed in the article.

**Keywords:** virtual reality; 4C/ID model; instructional design; competence acquisition; skills acquisition; vocational training; apprenticeship training

## 1. Introduction

Vocational training increasingly stresses a competence-based learning approach that requires comprehensive, action-oriented learning units with learning progress checks. Learners are confronted with authentic and complex real-life problems that require them to actively engage in their learning. These real-life problems help learners to integrate the competencies necessary for effective task performance in work settings [1,2]. Problems with the consistent implementation of the competence-based learning within training are common, for example, in the training of vehicle painters. Here, various techniques for applying a coat to a workpiece must be trained. Adequate, frequent, and action-oriented training, however, is hampered by economic (e.g., material costs), physical (e.g., environmentally sensitive materials), and social factors (e.g., limited support capacity). With Virtual Reality (VR), competence-based learning can be supported. With its features of immersion, interactivity, and imagination, VR can present a virtual environment that resembles the real world. It enables a high degree of authenticity of the learning situation, allowing learners to immerse themselves in a learning world where they can control their learning process and learn by exploring digital artifacts.

VR technology has become popular in recent years, and its effectiveness has been demonstrated in various educational settings [3–6]. In vocational training, VR has been widely adopted in various domains, including weld training, surgery, construction management, and military training, as a way to engage and motivate learners, decrease time to achieve skill mastery, cut down on material usage, and improve final performance outcomes. Although many studies reveal the effectiveness of VR across learning domains and contexts, the potential of VR technology for vocational training has not yet been explored in depth [7]. Studying this potential is important since VR can offer possibilities for

creating situated learning experiences in a safe, exploratory practice space as often found in vocational settings.

Moreover, VR technologies open new opportunities for effective training with low cost and fewer hazards, which compensates for the inherent drawback involved in traditional vocational training [8]. Hence, vehicle painting seems very suitable for the use of VR. For example, no haptic feedback from the 3D workpieces is required (apart from the spray gun), as these are not touched during application. In VR, feedback regarding coating thickness can be provided immediately, whereas in the real world, several hours are required until results become visible.

However, VR is merely a learning tool comparable with a book or a chalkboard. Learning tools by themselves do not teach, but instructional methods embedded in the tool presentation affect learning activities [9,10]. Instructional methods offer a set of practical procedures that consider several principles of human learning, specifically, the conditions under which learning occurs [11]. Therefore, to use VR as a learning tool, an appropriate instructional design model guiding the development of a VR-supported training environment is inevitable. Thus, the presented *HandLeVR* (The HandLeVR research project (01.01.19–28.02.22; https://handlevr.de/; accessed on 17 June 2022) is funded by the German Federal Minister of Education and Research (grant number: 01PV18002B) with partners from the University of Potsdam, the University of Duisburg-Essen, ZWH e.V., and Mercedes Benz Group AG (also known as Daimler AG). The software and accompanying materials are available as open-source under https://github.com/HandLeVR; accessed on 17 June 2022) (see supplementary materials) research project aims to develop and evaluate an effective VR training application using a highly validated instructional design model, namely the four-component instructional design (4C/ID) model, originally developed by [12]. The VR training system, referred to as the VR Painting Simulator, intends to train coat applications on 3D workpieces. It facilitates competence-based learning in the vocational training of vehicle painters in Germany, mainly for first-year students. So far, the training of vehicle painters has mainly consisted of the behavioral observations of trainers or more experienced trainees, memorization of action sequences, and a few practical painting exercises. These are rare because the quality of trainees' work is usually not sufficient for real customer orders. The VR Painting Simulator includes a set of learning tasks that differ from each other regarding various parameters (e.g., type of workpiece) as well as in terms of complexity [13]. It intends to help trainees to achieve the overarching goal of vocational action competence and to facilitate the integrative acquisition of competencies, i.e., knowledge, skills, and attitudes [14].

The *HandLeVR* project is fundamentally based on a three-part learning process. First, trainers define a learning task for one or more trainees. In the next step, this learning task is carried out by the trainees. Finally, the learning task is evaluated jointly by the trainer and the trainee. To illustrate this learning process, the training application is based on three components: an authoring tool, the VR Painting Simulator, and a reflection application. The three-part structure is illustrated in Figure 1. The trainer uses the authoring tool to create and assign ready-to-use learning tasks for trainees. In the VR Painting Simulator, trainees work on the assigned tasks. To do this, the trainees are immersed in a paint booth that has been recreated in detail and contains a 3D workpiece clamped on a painting stand (see Figure 2). Moreover, there is a monitor, a virtual trainer, and a poster to test the spray pattern. To move forward or backward, different colored coins are presented. In the dominant hand, the trainee holds a 3D-printed spray gun. Using replicated guns instead of commercial VR controllers was deemed acceptable to appropriately imitate the necessary action sequences (e.g., operating adjustment screws, pressing the lever) during painting and thus facilitate transfer to real-world painting settings. During coat application, the performance is recorded and stored along with a variety of parameters (e.g., layer thickness, distance between workpiece and spray gun). These data serve as the basis for evaluating performance in the reflection application. For a detailed description, please refer to [15].

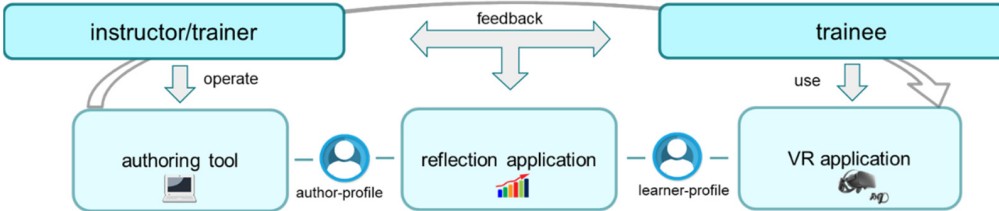

**Figure 1.** The three-part structure of the VR training application.

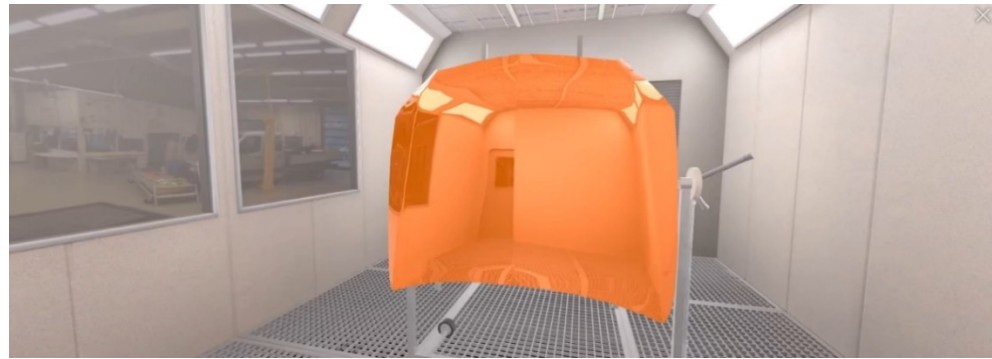

**Figure 2.** The paint booth.

Within this paper, we investigate whether the 4C/ID-model is applicable to VR and vocational training. Since instructional designs in VR are still missing, the research team aims to develop a training blueprint for VR. Hence, the paper is structured as follows. First, a brief overview of the underlying 4C/ID model and its components is given. Afterward, the design process along with the components to the field of vehicle painting and VR is described. The paper concludes with a feasibility check.

## 2. Theoretical Background: The Four-Component Instructional Design Model

While emerging technologies have fostered the development of VR learning applications, these solutions often lack a thorough instructional approach [16,17]. Consequently, some of these VR learning applications contribute little to competence acquisition. Neither are learning processes often mentioned nor do models or theories form the basis for VR learning scenarios [6,18–20]. Rather, research is technology-driven and often focuses on anecdotes, case studies, and demonstrations of technical prototypes. It has to be stated that there are hardly any specific models and theories for VR for the instructional preparation of VR learning applications. This could also be shown by two recent studies [21,22], both of which pointed out the lack of instructional theories that guide research as well as practice around the educational technology of VR. Recently, however, theoretical models of learning in VR are emerging [17,23,24]. These three theories focus less on technical aspects of the medium and more on the underlying learning processes.

Therefore, the VR training application presented here is based on a highly validated instructional design model, namely the **four-component instructional design (4C/ID) model by** Van Merriënboer and colleagues [12]. Other models, such as ADDIE [25], characterize the main phases in instructional design, whereas the 4C/ID model offers specific guidelines to develop educational programs that support the acquisition of complex skills. The model is acknowledged as one of the most effective instructional design models for designing powerful learning environments facilitating integrated competence acquisition [26–29]. Competencies are defined as complex cognitive skills, consisting of integrated sets of constituent skills with their underlying knowledge structures and attitudes [30]. The 4C/ID model has a strong foundation in research and has already been applied in various settings [31], such as teacher training [32], communication training [33], medical education [34,35], technical training [36,37], computer programming training [38], and

information problem solving [39]. The model states that educational programs can always be described in terms of four interrelated components (see Figure 3).

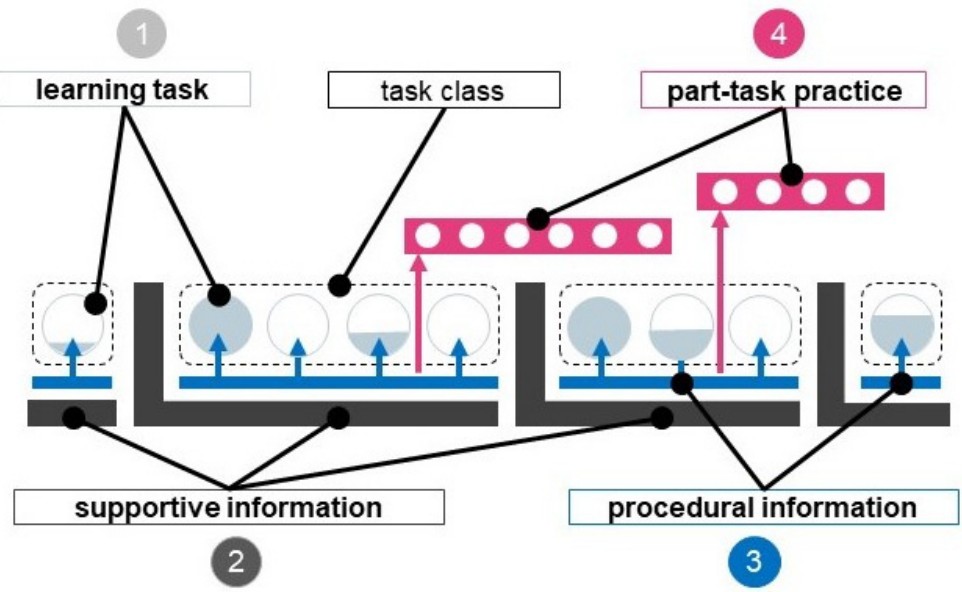

**Figure 3.** The 4C/ID model according to [12,13].

*Learning tasks* (first component) are concrete, authentic, and meaningful real-life experiences that confront learners with all constituent skills that make up a complex skill. In Figure 3, learning tasks are indicated by circles. The dotted rectangles around a set of learning tasks are task classes, which become more complex throughout the training program. Equivalent learning tasks belong to the same task class. The basic idea is to use a whole-task approach where the first task class refers to the simplest version of whole tasks in the real world. The final task class refers to the most complex situation that experts encounter in reality. In other words, if a task class presented at the start of training is too complex, it results in cognitive overload, which impairs learning and performance [40].

High variability of learning tasks within the same task class is important to ensure transfer of learning [41]. While there is no increasing difficulty for the tasks within one task class, they differ regarding the amount of support provided to learners. Learning tasks are sequenced from high to low support to avoid excessive cognitive load. Hence, learners receive much guidance and support working on the first learning task of a task class. Guidance and support typically decrease as learners acquire more expertise. This is presented by the filling of the circles (see Figure 3). Thereby, support is either product or process oriented. Product-oriented support focuses on providing the learner with assistance with the products involved in the training. Solution-process guidance focuses on assisting learners with the processes inherent to solving the learning task.

*Supportive information* (second component) provides the bridge between what learners already know and what they need to know to work on the learning tasks. Supportive information is assumed to be helpful for the learning and performance of non-recurrent aspects, which often involve problem solving and reasoning. This supports the learner in developing mental models and cognitive strategies. In Figure 3, supportive information is represented by the grey L-shaped figures. Because all learning tasks in the same task class require the same body of general knowledge, supportive information is not coupled to single learning tasks but task classes. Within a task class, it is always available to learners.

*Procedural information* (third component) is helpful for the learning and performance of recurrent aspects of the learning tasks—that is, aspects that are always performed in the same way and that can become routines. Because procedural information is relevant to the routine aspects of learning tasks, it is best presented exactly when needed to perform a task (i.e., just in time), after which it quickly fades for subsequent learning tasks. Procedural

information provides learners with the step-by-step knowledge they need to know to perform the recurrent skills. Procedural information can be provided as specific feedback just before, during, and shortly after learners practice the task. In Figure 3, procedural information is represented by blue rectangles with upward-pointing arrows.

*Part-task practice* (fourth component) consists of practice items for recurrent skills that need a very high level of automaticity. In Figure 3, part-task practice is represented by small pink series of circles, representing practice items. This promotes rule automation driven by the repeated practice of recurrent constituent skills. Part-task practice promotes the compilation of procedures or rules and especially their subsequent strengthening, which is a slow process that requires extensive training. In contrast to the more holistic learning tasks, such practices are extremely short, simplified, and with a sole focus on the specific skill. Part-task practice for a particular recurrent aspect of a task can begin after it has been introduced in a learning task, and then it is intertwined with the learning task to provide distributed practice.

According to the 4C/ID model, learning tasks, supportive information, procedural information, and part-task practice are crucial to design a learning environment. In the presented research, we applied the 4C/ID model to the field of vehicle painting. To the best of our knowledge, the model has never been used to create a learning environment in VR. Consequently, we aim to investigate the suitability of the 4C/ID model to design a VR training application for vehicle painters and describe the instructional design process step by step.

The following section describes the process of developing the VR Painting Simulator in detail. First, the *HandLeVR* project and its structures and methods are introduced. After that, the development steps towards a working and efficient learning environment are presented.

## 3. Design Process

### 3.1. Characteristics of the HandLeVR Project

Figure 4 illustrates the project process and Figure 5 gives insights into the instructional design process within *HandLeVR*. The project ran over a period of three years (2019–2021), and this paper was written during the project period. Therefore, the following results are preliminary. In 2021, final results (e.g., the VR Painting Simulator, accompanying materials as tutorials) were published as an open educational resource. Further information (e.g., reports, journal articles) is provided on our homepage. A video presenting a single learning task can be found on our homepage (https://www.youtube.com/watch?v=cQI045pJa1Y; accessed on 17 June 2022) as well.

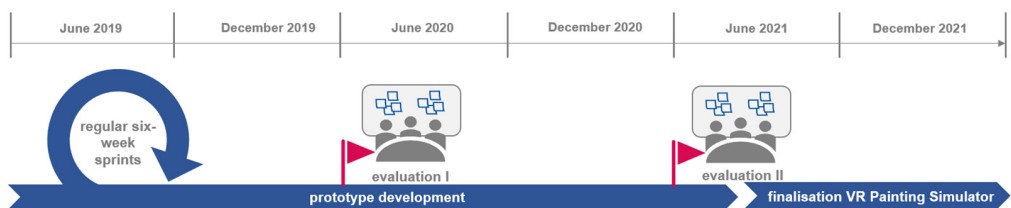

**Figure 4.** The *HandLeVR* project process.

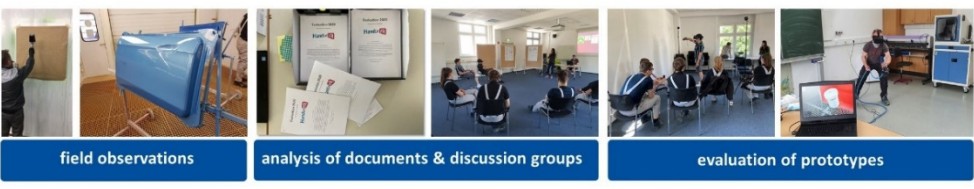

**Figure 5.** Instructional design process within the *HandLeVR* project.

In *HandLeVR*, we work in a collaborative team: computational scientists, instructional designers, and experts in vehicle painting. *Mercedes Benz GmbH*, a national training com-

pany in the field of vehicle painting, is part of the project. Therefore, we did not conduct a laboratory study but a field study.

In 2019, instructional design started by studying the training process. This included the analysis of trainers, trainees, job tasks, and customer orders as well as existing national curricula [42]. First, we conducted qualitative interviews with six aspirant vehicle painters and one trainer working for *Mercedes Benz GmbH* to gather first-hand information regarding the vocational training of vehicle painters. Unobtrusive observations of traditional teaching and learning scenarios provided further insights that are difficult to verbalize (e.g., trainer's supportive activities during the painting process). Additionally, the paint booth, the paint mixing room, and several workpieces were shown and explained in detail by the trainer. Hence, we collected plenty of information about the physical and psychological characteristics, learning lifestyle, and prior knowledge of the trainees (e.g., highly motivated, average age of 18, high affinity to technology). These findings were presented to and extended with three trainers from another national training center. Second, learning objectives were defined by studying the job tasks, customer orders, and national curricula, objectives, and required competencies a trainee should achieve after training becomes visible (e.g., layer thickness of 120 μm, personal protective equipment). As in many other vocational trainings, curricula in vehicle painting are based on modules with increasing complexity (e.g., preparing uncoated metal surfaces). This modular structure helped us to form the structure of the VR Painting Simulator.

The results of the analyses served as a basis for the technical requirements that our instructional designers defined for the VR Painting Simulator and that were implemented by our computer scientists.

Due to the rapid technological progress of VR technology as well as constantly changing requirements, the design process for the VR Painting Simulator needs to be highly dynamic and agile. Therefore, the project team follows the methodology of design-based research [43]. Since the requirements for the software already changed during the implementation, traditional approaches to software development (e.g., waterfall model) are no longer adequate. Hence, rapid prototyping was addressed by **Scrum** as an agile software development method [44]. Therefore, instructional and practical requirements are continuously transmitted to the development team and stored in a requirements collection (*Scrum*-Backlog). Every six weeks (*Scrum*-Sprint), intermediate implementations are presented, and working prototypes are practically tested and interdisciplinarily discussed within a review. This results in new requirements, and the development cycle begin anew. Numerous design iterations made it possible to improve results repeatedly. Next to agile project processes, two formative evaluations were implemented. In summer 2020, 14 trainees from *Mercedes Benz GmbH* tested typical learning tasks of the VR Painting Simulator. Questionnaires as well as discussion groups uncovered strengths and weaknesses (e.g., too much text to read). Taking user feedback into account, the prototypes were improved significantly afterwards. In summer and autumn 2021, the second evaluation was conducted.

Within the design process, several design failures became visible. First, the pressure tube that is usually held in the non-dominant hand was not taken into consideration when designing the VR Painting Simulator. The lack of the tube resulted in confusion among the trainees. Second, the paint booth and its elements are designed highly realistic and shall be presented in detail. Elements such as buttons or the floor are highly authentic. In terms of learning, this was not necessary. While some elements need to be designed in high fidelity to support learning adequately (e.g., the 3D paint gun should be identical to a real one in terms of haptics or weight), other elements can be represented more abstractly. Within the project team, there have been very different views on this issue. Computer scientists tend to have high demands in terms of fidelity and quality towards VR, whilst instructional designers tend to focus on developing learning scenarios. In this context, the didactical reduction is crucial due to cognitive processes [45]. Unnecessary cognitive load

may be induced through VR learning environments that try to resemble the real world at the highest quality level. Reduction to the learning relevant elements is recommended.

### 3.2. Design Steps along with the 4C/ID Model

To apply the 4C/ID model, we used a 10 step process—a prescriptive approach to the 4C/ID model that is practicable for instructional designers [13,46]. A detailed description of the four components of the 4C/ID model and the corresponding steps can be found in [13]. These 10 design activities guide instructional designers in performing analysis to uncover the required knowledge, skills, and attitudes and explain how to use these outcomes to form the subsequent training design according to 4C/ID principles. Each of the four components corresponds with a specific design step. In this way, the design of learning tasks corresponds with step 1 *Design Learning Tasks*, the design of supportive information with step 4 *Design Supportive Information*, the design of procedural information with step 7 *Design Procedural Information*, and the design of part-task practice with step 10 *Design Part-Task-Practice*. These four are the main steps. The other six stepsare supplementary and are performed when necessary. These steps include (2) *Sequence Task Classes*, (3) *Set Performance Objectives*, (5) *Analyze Cognitive Strategies*, (6) *Analyze Mental Models*, (8) *Analyze Cognitive Rules*, and (9) *Analyze Prerequisite Knowledge*. To help the reader keep a coherent overview, the 10 steps are reviewed in four sections—one for each component. Thus, the next section discusses the application of the 4C/ID model in vocational training in the field of vehicle painting along with the model principles.

*Learning tasks:* Three easy-to-difficult task classes are identified and validated in collaboration with several experienced trainers and teachers in the field of vehicle painting: (1) new part painting, (2) refinish painting, and (3) spot-repair painting. The learning tasks are all conducted in VR, which allows trainees to work on authentic real-life tasks in a computer-simulated learning environment. In *HandLeVR*, we aim to develop a high-fidelity virtual learning environment using VR helmets and 3D-printed spray guns. Trainees should easily immerse themselves in the learning scenario.

Below, we focus on the first task class, new part painting, which contains six learning tasks. In subsequent task classes (e.g., refinish painting), trainees work on increasingly complex tasks that require more knowledge (e.g., spot repair: masking and removing tape) or more embellished knowledge for effective performance than the preceding and easier task classes (step two). Each learning task should offer whole-task practice, confronting the learner with the constituent skills important for performing the task, including their associated knowledge and attitudes. The learning tasks must differ from each other in all dimensions that also differ in reality, but they are equivalent within a task class concerning complexity and required knowledge. Transferred to vehicle painting, a learning task covers a customer order. Learning tasks vary regarding workpieces as well as the type and number of coatings. This variability of practice contributes to the transfer of competencies to other non-familiar situations. Most learning tasks include both routine and non-routine aspects. Routine aspects are, for example, preparing the workpiece for coating and pre-checking the spray pattern. Non-routine aspects are actions that are rather new to learners and require effort. For example, combining information from various sources (e.g., customer order, technical instructions, environmental temperature) is a non-recurrent skill that requires the combination of cognitive schemata to engage in problem-solving behavior and the application of cognitive strategies [46]. Besides varying in their content and presentation, tasks may also vary in their instruction. Hence, there is a decrease in support and guidance as the trainee proceeds through the task class. This principle is called **Scaffolding**. In Figure 3, the principle is illustrated by the filling of the circles. Product-oriented support is provided by different types of learning tasks. At the beginning of a task class, case studies confront the trainee with a given state, the desired goal state, and a solution. Within completion tasks, which appear at the end of a task class, trainees are asked to fulfill increasingly larger parts of a given incomplete solution or customer order (e.g., apply two of three coats). Solution-process support includes, for example, fully reasoned best-practice

modeling examples of higher training years or detailed step-by-step instructions of the trainer. Finally, conventional learning tasks confront trainees with a typical customer order. Along with a given state and the desired goal state, the trainee has to complete the customer order without any support and guidance.

As in any other competency-based curricula, assessment instruments are crucial elements in the process of complex learning (step three). They assess whether predefined standards or objectives have been met. Performance standards (e.g., layer thickness) provide trainees with feedback to improve and guide their learning process as well as inform trainers about the actual performance and the desired outcomes. In the first task class, new part painting, trainees are prompted to enter the correct distance between the workpiece and spray gun. One way to inform users about the performance standards linked to the desired professional behavior is to use worked examples. In *HandLeVR*, a worked example can take the form of a VR recording of an experienced trainer applying a coat on a specific workpiece and simultaneously explaining each step. Therefore, a worked example can become a modeling task where not only the solution is offered but also the process to come to the solution. Another way to inform users about the performance standards contains providing corrective feedback. In *HandLeVR*, trainees receive visual feedback by illustrating layer thickness on a so-called Heatmap on the workpiece. Different colors indicate layer thickness (see Figure 6). Number-based performance parameters (e.g., wasted coat in µm) are also presented to trainees after each task. Additionally, after completing learning tasks, learners are instructed to rate their performance by deciding how many of three golden spray guns correspond to their performance.

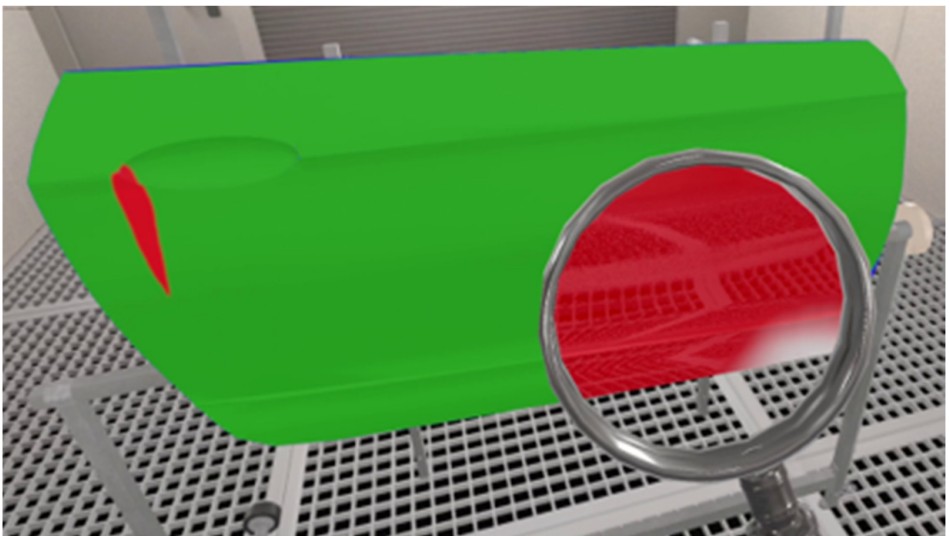

**Figure 6.** Supportive information: layer thickness (*Heatmap*).

To work on learning tasks, trainees enter the paint booth and receive a task overview. This overview is presented on a monitor. Trainees are informed which learning tasks are fulfilled and which are outstanding (see Figure 7). Within the first learning task, the trainee is prompted with coating a mudguard. The virtual trainer is highly supportive, providing additional information about safety at work, drying periods, and measuring instruments. An experienced trainer sets a best-practice-example, applying coat on a mudguard and simultaneously explaining his actions step by step. After observing and checking for the spray pattern, the trainee is next to apply an orange one-layer coat on a new mudguard.

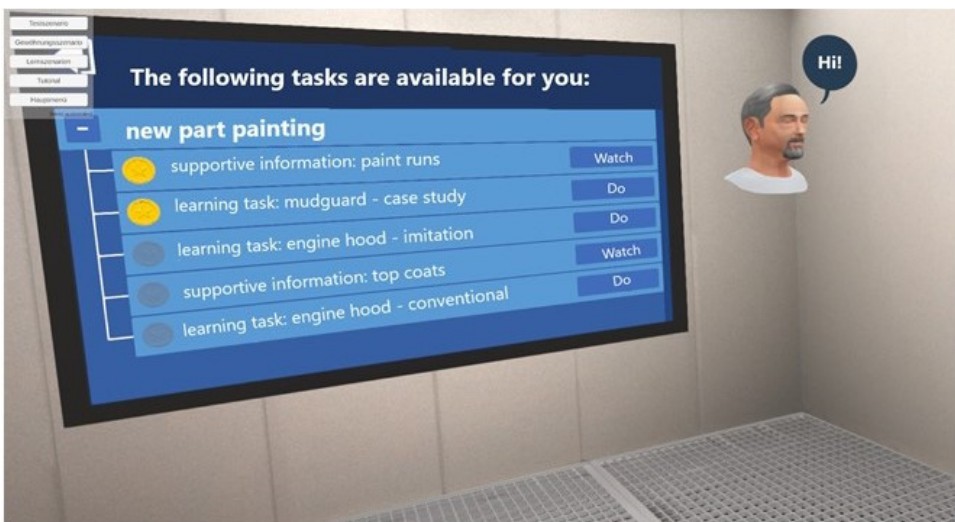

**Figure 7.** Overview of the task class new part painting.

*Supportive information:* Once learning tasks and assessment instruments are developed and sequenced, further steps focus on the identification and development of information that supports the learning process. Steps four to six entail the design and development of supportive information. Supportive information is tailored to the content of each task class and therefore available throughout all tasks of a task class. It provides the trainee with new information that is necessary for the fruitful completion of non-recurrent and non-routine aspects related to problem-solving, reasoning, and decision making while completing the learning tasks. Its function is to facilitate schema construction such that trainees can deeply process new information, in particular by connecting it to already existing schemas via elaboration. Designing supportive information requires the analysis of cognitive strategies and mental models (steps five and six). Cognitive strategies are those that proficient task-performers use to solve problems, such as heuristics and rules-of-thumb. Mental models reflect how the learning domain is organized. They can be described as personal theories that are actively created and are informed by previous experiences with similar situations.

Interviews with trainers and teachers in the field of vehicle painting reveal that trainees, especially at the beginning of training, need more product-oriented as well as process-oriented support (see Section 2). They need practice to adopt rules-of-thumb (e.g., 15 cm distance between workpiece and spray gun) and theoretical concepts (e.g., personal protective equipment). Therefore, various rules-of-thumb and theoretical concepts are collected and assigned to task classes. The first task class comprises supportive information focusing on (1) cognitive strategies, such as maintaining the distance between the spray gun and workpiece and applying coat in the right sequence, and (2) mental models regarding work plans, paint runs, spray pattern, and top-coats. Supportive information is provided by short video clips, slide shows, drag and drop, multiple-choice tasks (see Figure 8), or explanations by the virtual trainer. Doing this, we take multimedia principles into account to ensure that trainees work in an environment that is goal-effective and meaningful but not overloading [40]. Next to the provision of cognitive strategies and mental models, supportive information also encompasses cognitive feedback. This type of feedback assists learners in reflecting on the quality of their problem-solving processes and solutions. As mentioned in the section before, feedback comprises several parameters (e.g., layer thickness, time needed) illustrated as numbers or on the workpiece itself.

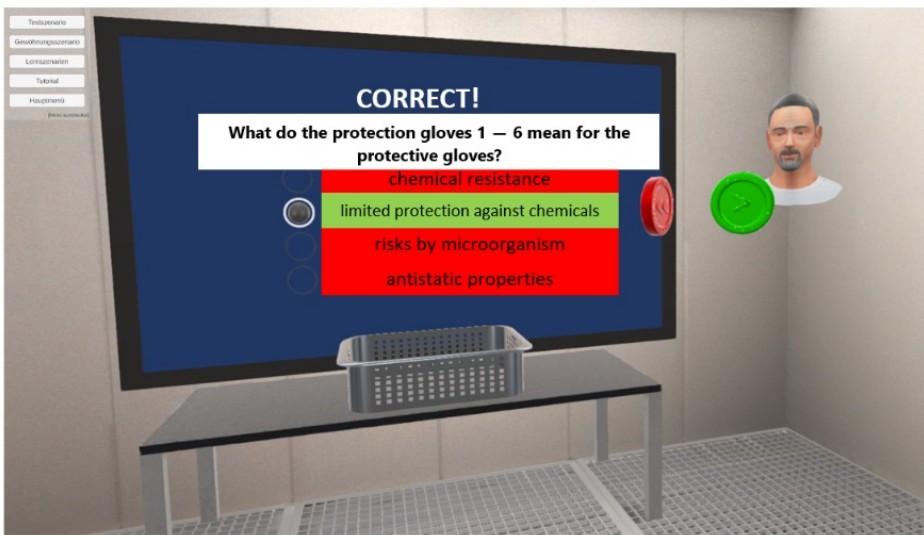

**Figure 8.** Supportive information: personal protective equipment.

*Procedural information:* Whereas supportive information pertains to nonrecurrent aspects, procedural information pertains to recurrent aspects. It specifies exactly how to perform these aspects and is preferably presented precisely when trainees need it during their work on the learning tasks. For subsequent learning tasks, this procedural information quickly disappears. This principle is called **Fading**. Procedural information is presented in short units because only the presentation of relatively small amounts of new information at the same time can prevent processing overload during practice.

As mentioned in steps seven to nine, designing procedural information requires the study of cognitive rules and prerequisite knowledge. The analysis of cognitive rules identifies condition–action pairs. In the field of vehicle painting, a rule, for example, could be "If you are too close to the workpiece, you apply too much coat, and as a result, paint runs appear". The analysis of prerequisite knowledge identifies what learners need to know to correctly apply those condition–action pairs. Here, a trainee needs to know the right distance between the spray gun and workpiece, which is about 15 cm. VR can provide procedural information just in time so that the learner can start applying without referring to, for example, a manual.

In the first task class, we support trainees in maintaining the correct distance by depicting a colored beam (see Figure 9). Three colored areas are displayed on the beam: red means too close, blue too far away, and green adequate. Trainers in the field of vehicle painting confirm that maintaining distance while applying coat is a huge issue, not only for inexperienced trainees but also for proficient task-performers. During the design process, we tested several options to support trainees in maintaining the right distance. Based on trainees' and trainers' feedback, the beam could prevail over other proposed solutions (e.g., arrows).

*Part-task practice:* Often, learning tasks provide enough of an opportunity to practice both nonrecurrent and recurrent aspects. However, additional practice is sometimes necessary for selected recurrent aspects of a complex skill to develop a very high level of automaticity. As experts in the field of vehicle painting mentioned, maintaining distance while applying a coat needs a huge amount of repetition. Therefore, we add short practice items with simplified rectangular workpieces. The colored beam is integrated. Part-task practice is best intertwined with the learning tasks and therefore implemented three times within the first task class.

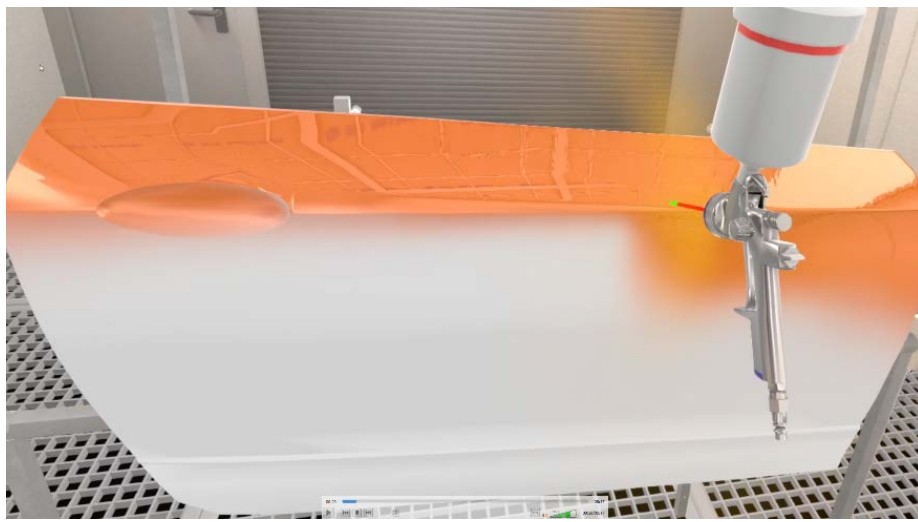

**Figure 9.** Procedural information: a beam indicating the distance between the workpiece and the spray gun.

Figure 10 and Table 1 provide additional material for the design process. Figure 10 is based on [12,13,46].

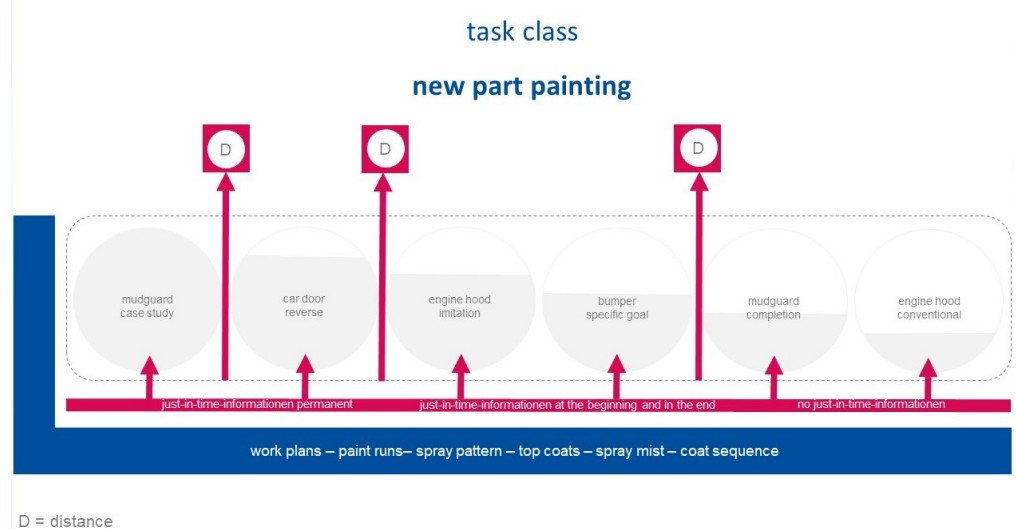

D = distance

**Figure 10.** First task class: new part painting.

Further information is exhibited in Table 1. It presents a simplified training blueprint for the field of vehicle painting. The table structure was originally developed by van Merriënboer and colleagues [46] and transferred to the field of vehicle painting. The table illustrates the four model components implemented in the first learning task. First, three out of six learning tasks are briefly described. These learning tasks show a decrease in support and guidance. Second, supportive information is specified for the task class. Conceptual models, especially regarding preparation and follow-up of a coating, and cognitive strategies, for example, rules-of-thumb of how to apply a coat, are named. Moreover, at the end of each learning task, trainees receive feedback on their work. Third, procedural information (i.e., the beam that indicates the adequate distance between workpiece and spray gun) is specified for each learning task where it is relevant. Fourth and last, is the program describes where to initiate additional part-task practice. In this blueprint, it is assumed that trainees receive three additional practice items.

**Table 1.** A simplified training blueprint.

| | | |
|---|---|---|
| *task class 1: new part painting* <br><br> Trainees are confronted with customer orders belonging to new parts of a vehicle. The orders differ regarding learner support, type of workpiece, type of coat, number of coatings etc. No refinishing and no spot repair has to be done. | | |
| *supportive Information: cognitive strategies* <br><br> - systematic approach to problem solving (SAP) of reading and understanding producer-dependent processing guidelines <br> - SAP/rules-of-thumb for applying coat regarding sequence, distance, angle etc. <br><br> *supportive Information: mental models* <br><br> - conceptual model of protective gloves <br> - conceptual model of drying times depending on various context factors (e.g., temperature) <br> - conceptual model of amount of coat depending on type of workpiece <br> - conceptual model of measuring instruments depending on material (e.g., steel sheet) <br> - conceptual model of components and structure of a two-layer-coating <br> - conceptual model of spray pattern (errors, their causes and remedies) | | |
| *Learning task 1.1: case study* <br><br> Trainees receive a worked-out example (best-practice of a virtual trainer) of applying a blue coat on a mudguard. After this, they have to do an orange one-layer coating on another mudboard. | *Procedural information:* <br><br> - beam that indicate the correct distance between spray gun and workpiece (permanently active) | *Part-task-practice:* <br><br> - applying coat on simplified rectangular workpieces (3x with permanent, semi-permanent and no procedural information) |
| *Learning task 1.3: imitation* <br><br> Trainees receive another worked-out example (best-practice of a virtual trainer) of applying a coat on an engine hood. They should focus on trainers' motoric skills. After this, they have to do an one-layer coating on another engine hood. | *Procedural information:* <br><br> - beam that indicate the correct distance between spray gun and workpiece (active at the beginning and in the end) | |
| *Learning task 1.6: conventional* <br><br> Trainees receive a customer order. They have to apply three coats on an engine hood. After this, they have to rate their performance on the basis of one to three golden spray guns. | | |
| *Supportive Information: corrective feedback* <br><br> - trainees receive feedback in learning tasks 1.1, 1.3 and 1.6. <br> (1) a colored map on the workpiece indicate layer thickness <br> (2) numerous parameters (e.g., wasted coat in %) | | |

## 4. Conclusions

The design process—lasting more than two years—has revealed so far that the 4C/ID model generally fosters the instructional design process of an action-oriented VR training application in the field of vehicle painting. This is an important finding as instructional designs for VR are still lacking [17,19,47].

The 4C/ID model provides a structured approach for designing a VR training application in vocational education. The model components, the model principles, and the 10 step approach enable a systematic and step-by-step design process that fits *HandLeVR*'s dynamic project structure. Therefore, we recommend the 4C/ID model to agilely develop vocational training applications in VR. According to our experience, it seems to be a suitable approach for further VR applications in the field of vocational training. This is due to several reasons. First, the model departs from whole tasks. These tasks are crucial for the integration of knowledge, skills, and attitudes. This integration is necessary to enable transfer from educational settings to professional settings and as a result to achieve the overarching goal of vocational action competence. Second, the 4C/ID model differs between theory and practice, which is a common distinction in vocational education. The information that supports learning processes during working on the learning tasks is presented to learners before or just in time when needed. The additional information that is necessary to perform the tasks in a task class with a higher level of complexity is presented in the next task class, so that practice model is more learner-centered, depends on constructivist learning approaches, is based on cognitive load theory, and has already been applied in various settings [31]. Moreover, an analysis by [36] reveals that the 4C/ID model is highly effective in terms of complex learning. For these reasons, we selected the 4C/ID model for use in our project. Consequently, we do not provide any information regarding the effectiveness of the 4C/ID model compared to other, more traditional approaches.

Even though the 4C/ID model seems to significantly foster design building, there were some implementation and methodological issues. During the design process, we encountered several challenges. First, we recognized that implementing each model component to VR is somehow difficult. VR helmets can only be worn for a limited time, approximately 30 min, without limitations (e.g., headaches, dizziness) [48]. Outsourcing certain model components (e.g., supportive information) to the real world may reduce the cognitive load of wearing VR helmets but may compromise integrated competence acquisition. Second, presenting textual information in VR is challenging. In vehicle painting, work plans, technical datasheets, etc. are often wordy, and only a fraction of information is relevant to task execution. We had to prepare new document versions to avoid cognitive overload. Thus, we had to delete irrelevant content, and the new documents are no longer comparable to those in the real world. Third, during the design process, we recognized that the preparation and follow-up of a coating are crucial to produce adequate results. A paint booth as a single room in VR is not adequate to display all competence-relevant learning scenarios. A rudimentary version of a paint mixing room was developed but is not yet part of the of the learning tasks.

Next to implementation and methodological issues, using VR as a learning tool is certainly worth discussing. VR can present a virtual environment, here a paint booth, that resembles the real world, allowing learners to immerse themselves in this world. This enables the training of vehicle painting through repeated unsupervised practice without risks and material wastage but with automated feedback and support [8]. Otherwise, we have no evidence supporting the direct translation between painting performance in VR and real life. An adequate painting in VR cannot guarantee similar results in reality. Thus, some experienced trainers only achieved mediocre results in VR, whereas others performed well. High interindividual variability between trainers regarding their performances became visible. We can conclude that VR learning environments may not replace real-life training but augment vocational training, especially when support capacity is limited and real-life practice is rare. It may provide training opportunities which may consequently result in enhanced performance in the real world. Moreover, it is noteworthy that the VR Painting

Simulator is adaptable. Learning tasks and support can be integrated or faded based on the needs and competencies of an individual trainee.

As a result, we need to specify the scope of the VR Painting Simulator. We state that the VR Painting Simulator is an effective tool to support integrated competence acquisition, but it is still only a useful addition and not a substitute or a stand-alone solution. Traditional learning scenarios within vocational training (e.g., observing the trainer while listening to his or her explanations of the trainer) are still needed.

Taken together, we have applied the highly validated 4C/ID model in the field of vehicle painting using VR technology. According to our experiences, the 4C/ID model seems to be suitable to design VR learning environments fostering competence acquisition. We are pleased about the new insights and the lessons learned, and we are already looking forward to reporting the results of the second formative evaluation with 50 to 60 aspirant vehicle painters. Future studies on the VR Painting Simulator will address constructs such as presence and cognitive load but also the gap between VR skills and skills in real-world scenarios.

**Supplementary Materials:** Supporting information can be downloaded at https://handlevr.de or https://github.com/HandLeVR (accessed on 15 May 2022).

**Funding:** The *HandLeVR* research project (1 January 2019–28 February 22; https://handlevr.de/, accessed on 16 May 2022) is funded by the German Federal Ministry of Education and Research (grant number: 01PV18002B) with partners from the University of Potsdam, the University of Duisburg-Essen, ZWH e.V., and Mercedes Benz Group AG (also known as Daimler AG).

**Institutional Review Board Statement:** Not applicable.

**Informed Consent Statement:** Not applicable.

**Data Availability Statement:** Not applicable.

**Conflicts of Interest:** The author declares no conflict of interest.

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
