# Peer review of "Vocational Training in Virtual Reality: A Case Study Using the 4C/ID Model"

_mti, doi:10.3390/mti6070049_

Round 1

Reviewer 1 Report

The manuscript focuses on VR in VET a timely and significant topic. It is a well-written study implementing a validated instructional model. Its main weakness is the absence of any user evaluation data to support statements concerning its application results (e.g. L14-16).

Moreover, the following issues should be addressed:

Abstract: References in an abstract are rarely encountered. Also, additional key components and highlights of the study should be included such as research method, main findings and conclusions.

Theoretical background: L93-94: This statement presents a rather rigid approach. It would be advisable to refer to systematic reviews that have explored instructional design for educational interventions in VR, e.g. (Kim et al., 2019; Pellas et al., 2021). In this context, another model worth mentioning is (Makransky & Petersen, 2021).

L128-129: It is not clear how data sheets are related to the products of a learning procedure.

L149: The part-task practice is not conceptually clear to the reader in comparison with learning tasks. Please highlight their differences and consider adding an example.

Design: This section spans over 7 pages and should be divided at least into two subsections, one dealing with the Design Process (Materials) and the second on Evaluation Methods and Procedures.

L253: Improve the coloring of rows in Table 1 to illustrate clearly which steps correspond to each model components.

L254: The word “Finally” is confusing as it is the beginning of this descriptive section on learning tasks. Text appears to be missing.

References

Kim, Y. M., Rhiu, I., & Yun, M. H. (2019). A Systematic Review of a Virtual Reality System from the Perspective of User Experience. International Journal of Human–Computer Interaction, 1–18. https://doi.org/10.1080/10447318.2019.1699746

Makransky, G., & Petersen, G. B. (2021). The Cognitive Affective Model of Immersive Learning (CAMIL): a Theoretical Research-Based Model of Learning in Immersive Virtual Reality. Educational Psychology Review. https://doi.org/10.1007/s10648-020-09586-2

Pellas, N., Mystakidis, S., & Kazanidis, I. (2021). Immersive Virtual Reality in K-12 and Higher Education: A systematic review of the last decade scientific literature. Virtual Reality, 25(3), 835–861. https://doi.org/10.1007/s10055-020-00489-9 

Reviewer 2 Report

The author presented some interesting work on designing VR-enabled training for vehicle painting by applying the 4C/ID model. The paper is well written in general and is easy to follow. More importantly, the author investigated an important but often omitted topic regarding VR-enabled training and learning – how to design effective learning by utilising the affordances of VR. The followings are a few recommendations the author may wish to address.

(1)    The author may wish to provide some background information on the conventional ways for training vocational competencies in the field of vehicle painting and better state why she thinks VR can be a promising tool before moving on to the actual implementation of the project.

(2)    There could be a more in-depth review of prior works on the area of using VR for vocational training or even immersive learning in general. I would recommend the author have a section dedicated to related work.

(3)    The technical details of the VR system can be further amended by the author. I’m not sure if trainees are supposed to use off-the-shelf VR controllers when learning the techniques in HandLeVR or a special device that mimics the actual painting device has been augmented in the VR scenarios. If it’s the former, the author may need to better discuss the generalisation of the skills and knowledge learned in VR to real-world tools and environments.

(4)    It would be interesting to see if any future works will be carried out in the conclusion section.

Round 2

Reviewer 1 Report

The author addressed all identified issues except one in a satisfactory way.